# Intense Blue Photo Emissive Carbon Dots Prepared through Pyrolytic Processing of Ligno-Cellulosic Wastes

**DOI:** 10.3390/nano13010131

**Published:** 2022-12-27

**Authors:** Loredana Stan, Irina Volf, Corneliu S. Stan, Cristina Albu, Adina Coroaba, Laura E. Ursu, Marcel Popa

**Affiliations:** 1Faculty of Chemical Engineering and Environmental Protection, “Gheorghe Asachi“ Technical University, D. Mangeron 73 Ave., 700050 Iasi, Romania; 2Centre of Advanced Research in Bionanoconjugates and Biopolymers, “Petru Poni” Institute of Macromolecular Chemistry of Romanian Academy, Grigore Ghica Voda 41A Alley, 700487 Iasi, Romania; 3Academy of Romanian Scientists, Ilfov Street, 077160 Bucharest, Romania

**Keywords:** carbon dots, pyrolysis, photoluminescent materials

## Abstract

In this work, Carbon Dots with intense blue photo-luminescent emission were prepared through a pyrolytic processing of forestry ligno-cellulosic waste. The preparation path is simple and straightforward, mainly consisting of drying and fine grinding of the ligno-cellulosic waste followed by thermal exposure and dispersion in water. The prepared Carbon Dots presented characteristic excitation wavelength dependent emission peaks ranging within 438–473 nm and a remarkable 28% quantum yield achieved at 350 nm excitation wavelength. Morpho-structural investigations of the prepared Carbon Dots were performed through EDX, FT-IR, Raman, DLS, XRD, and HR-SEM while absolute PLQY, steady state, and lifetime fluorescence were used to highlight their luminescence properties. Due to the wide availability of this type of ligno-cellulosic waste, an easy processing procedure achieved photo-luminescent properties, and the prepared Carbon Dots could be an interesting approach for various applications ranging from sensors, contrast agents for biology investigations, to photonic conversion mediums in various optoelectronic devices. Additionally, their biocompatibility and waste valorization in new materials might be equally good arguments in their favor, bringing a truly “green” approach.

## 1. Introduction

Carbon Dots (CDs) are a new class of nanostructured materials that are starting to raise a growing practical interest due to their remarkable features, such as photoluminescent emission with high quantum yield (PLQY) conversion, lack of toxicity, high physical-chemical stability, surface functionalization potential, dispersibility in various solvents, and various preparation paths [1,2,3,4]. The most remarkable feature of CDs is their photoluminescent emission, with peaks typically located in the range of 410–520 nm depending on the excitation radiation wavelength [5]. This emission peak dependence is rather a less-common behavior among typical fluorescent compounds, being most probably a result of the specific mechanisms which govern their radiative processes [6]. The structural configuration of CDs consists in a defect rich graphitic core in 2–30 nm range, mainly composed of carbon atoms in sp^2^ hybridization. Various terminal functional groups (-C=O, NH, COOH, -OH) are present within the graphitic structure or attached on the core surface, having an essential role in the photonic radiative processes [7,8]. This structure ensures a notable physico-chemical stability, thus allowing the use of CDs in various dispersion media such as organic solvents, monomers, or polymer matrices [9,10,11]. Common synthesis procedures of CDs can be grouped as physical [12,13] and chemical methods, the latter being preferred due to the simplicity and quality of the resulting materials. The chemical processes mainly consist of thermal treatment [14,15,16] or oxidation in acidic environments [17]; in certain reported works, the process is ultrasonic or microwave-assisted [18]. Among them, the pyrolytic processing of the precursors might be seen as the most straightforward and technically unchallenging preparation method capable of yielding high-quality CDs.

In recent years, there has been a significant increase of interest in obtaining new nano/microstructured materials from biomass waste, targeting both preparation methods and implementation in various applications such as medical imaging or anti-tumor treatments [19], sensors, optical devices [20], catalysis, and wastewater treatment [21,22]. Thus, CDs were obtained by thermal processing of different feedstocks [23,24,25,26]. Activated carbon microstructures and carbon nanotubes were prepared by using ligno-cellulosic biomass waste from Miscanthus giganteus, which was treated with phosphoric acid and then pyrolytically processed at 500 °C [27]. In another approach, carbon nanotubes were obtained by thermal processing under CO, CH_4_ atmosphere, resulting from the gasification of biomass waste [28]. Carbon-based porous materials for electromagnetic radiation shielding have also been prepared from biomass waste [29], applications for this field being promising in limiting the effects generated by electromagnetic pollution. The pyrolytic carbonization of biomass wastes in the presence of potassium hydroxide allowed obtaining carbon nano/microstructures with high porosity and surface area (approx. 3560 m^2^/g), thus opening very promising perspectives in obtaining electrodes for electrochemical capacitors with a 170% higher specific capacitance than those currently used [30]. CDs with high intensity of photoluminescent emission were prepared from biomass residues with a high ligno-cellulosic content by hydrothermal extraction at 120–160 °C followed by carbonization of the products at 220 °C in a pressurized reactor. The resultant carbon nanostructures can be used for temperature sensors [31]. The use of activated carbon materials with a mesoporous structure, prepared from biomass with ligno-cellulosic content, open new perspectives for applications as catalysts support, usable in photocatalytic processes [32], while CDs prepared from various precursors (including natural sources) become increasingly interesting for electro-catalytic production of hydrogen [33].

In this work, Carbon Dots (CDs) were prepared through pyrolytic processing of forestry ligno-cellulosic waste. The resulting CDs are strongly photoluminescent in the blue region of the visible spectrum with characteristic excitation wavelength dependent emission peaks in 438–473 nm range and a remarkable 28% quantum yield in a relatively wide excitation range (350–390 nm), being easily dispersible in various solvents including water, ethylic alcohol, and acetone.

## 2. Materials and Methods

### 2.1. Materials

The ligno-cellulosic material used as precursor consists of spruce bark and other wood waste in the form of sawdust derived from the primary processing of the logs. The ligno-cellulosic waste was dried for 48 h at 80–90 °C and kept in sealed containers to avoid water retention. Ultra-pure distilled water (Millipore-Direct Q—Merk Milipore, Burlington, MA, USA) was used within all the preparation stages.

### 2.2. Methods

The Infrared (FT-IR) spectra were recorded in the 400–4000 cm^−1^ range using a Shimadzu IR Affinity 1S spectrometer (Shimadzu Corp., Kyoto, Japan) according to the KBr method. The thermal stability was studied on a Mettler Toledo TGA-SDTA 851e (Mettler Toledo, Greifensee, Switzerland) under an airflow rate of 20 mL/min. The heating rate was adjusted to 10 °C/min in the 50–800 °C range. The EDX investigation was performed on a Verios G4 UC Scanning Electron Microscope equipped with an EDS, EDAX Octane Elite energy dispersive spectrometer (Thermo Fisher Scientific, Waltham, MA, USA). The Raman spectroscopy was performed in the 100–3200 cm^−1^ range with a Renishaw inVia Reflex confocal microscope (Renishaw, Wharton, UK) equipped with a He–Ne laser at 633 nm (17 mW) and a CCD detector coupled to a Leica DM 2500M microscope (Leica Microsystems, Germany). All measurements were made in backscattering geometry using a 50× objective with a numerical (NA) value of 0.75. The HR-SEM micrographs were recorded with a Carl Zeiss NEON 40EsB Cross Beam System (Zeiss Microscopy, Jena, Germany) with thermal Schottky field emission emitter and accelerated Ga ions column. The samples were deposited on the analysis pad from a freshly prepared aqueous dispersion of CDs. X-ray diffraction (XRD) analysis was performed on a Rigaku Miniflex 600 diffractometer (Rigaku Corp., Tokyo, Japan) using CuKα-emission in the angular range 2°–55° (2θ) with a scanning step of 0.0025° and a recording rate of 1°/min. Emission spectra and the absolute photoluminescence quantum yields (PLQY) were recorded using a Horiba Fluoromax 4P fluorescence spectrophotometer equipped with Quanta-φ integration sphere (Horiba Ltd. Kyoto, Japan). The PLQY measurements were performed according to the equipment manufacturer’s procedure, the values being calculated by the provided Fluoressence software. Excited states lifetimes were investigated on an Edinburgh Instruments FLS980 photoluminescence spectrometer (Edinburgh Instruments Ltd. Livingston, UK). The double exponential function which was used to fit the fluorescence lifetimes of the samples is given by the equation:$$I\left(t\right)=I_{1}\cdot e^{\left(-t/\tau_{1}\right)}+I_{2}\cdot e^{\left(-t/\tau_{2}\right)}+I_{3}\cdot e^{\left(-t/\tau_{3}\right)}$$
where $\tau_{1}\;$− $\tau_{3}$ are the characteristic lifetimes and $I_{1}$ − $I_{3}$ are their relative amplitudes.

### 2.3. Preparation of the CDs Derived from Forestry Lingo-Cellulosic Waste

In Figure 1 are presented the preparation stages of the CDs prepared from ligno-cellulosic waste. The initially prepared waste according to Section 2.1. was additionally dried under vacuum at 100–120 °C for 24 h and then finely grinded to an average 100–150 μ size. The fine-grinding stage is essential for obtaining CDs with intense PL emission. In the next stage, the fine-grained ligno-cellulosic waste is thermally processed in a setup consisting of a quartz tube, a heating mantle, a temperature/flow controlled hot air source, and a cold-water container for the primary dispersion step [34].

In a typical preparation procedure, 0.3–0.4 g fine grinded ligno-cellulosic waste was added in the quartz tube and heated at 600 °C for 10 min. Following the completion of the thermal exposure stage, the pyrolysis product was immediately flooded with cold water (3–5 °C). The resulted CDs primary aqueous dispersion was evacuated and further centrifuged at 15,000 RPM for about 10 min. The collected aqueous supernatant with a clear transparent aspect contains dimensionally selected CDs. The cold-water flooding stage is critical for obtaining CDs with intense PL emission, due to the major influence over their final morpho-structural configuration and further over the PL emission intensity.

## 3. Results

As mentioned above, the structural configuration of CDs consists of a defect-rich graphitic core, mainly composed of carbon atoms in sp^2^ hybridization and various terminal functional groups inserted within the graphitic structure or attached on the core surface. As will be further presented, this configuration has an essential role in the photonic radiative processes which lead to their photoluminescent emission. The performed morpho-structural investigations aimed to confirm their structural configuration and also to provide information regarding their dimensional characteristics along with an in-depth evaluation of their PL properties.

### 3.1. EDX Investigation

The overall composition of the forestry ligno-cellulosic waste is presented in Table 1. A possible reason of iron and nickel presence might be a result of various primary processing operations of the wood (for ex. chain sawing, log cutting) or according to some studies [35,36] a result of natural assimilation by roots as essential micronutrients and from the atmospheric dust settled over time on the outer tree parts.

### 3.2. Thermal Analysis

The thermal analysis was performed both for a ligno-cellulosic waste sample (after the oven-drying stage) prior to pyrolysis and for the CDs resulting from the pyrolytic processing. CDs sample preparation consisted of freeze drying of the aqueous dispersion, resulting in a fine powder which was the subject of thermal behavior investigation. The mass variation diagrams recorded for the two samples are presented in Figure 2. In the case of the ligno-cellulosic waste sample, there was a first stage located within the 25−130 °C range where the mass loss was due to the elimination of the remnant moisture.

The significant mass loss (over 50%) occurred in the 250–400 °C range due to the advanced destructuration accompanied by volatiles elimination. In the last stage within 400–700 °C range, the processes of advanced destructuration went further, but the recorded mass variation was approx. 7–9%. In the case of the prepared CDs, the evolution of mass loss is predictable, being significantly reduced compared with the unprocessed ligno-cellulosic waste sample due to the partial pyrolytic process in which most of the destructuring processes are accompanied by volatiles exhaustion. Nonetheless, the mass loss is only partial within the thermal exposure stage which allows both the configuration of the carbonaceous and also maintain an optimal concentration of various functional groups. In the first decomposition stage, the mass variation was similar to that recorded in the case of the biomass sample, most probably due to the loss of remnant water. In the 200–400 °C range the mass loss was low (approx. 5%), while in the upper 400–700 °C range, the mass loss was moderately high (approx. 25%) as a result of advanced destructuration, which most likely involves the near-complete loss of the functional groups attached or contained within the graphitic core, which were still present after the pyrolytic processing with the thermal exposure parameters presented in Section 2.3.

### 3.3. FT-IR Analysis

The FT-IR investigation was performed for both the ligno-cellulosic waste sample and for the CDs resulting from the pyrolytic processing. The recorded spectra are presented in Figure 3a,b.

Table 2 details the recorded significant peaks and their assignment for the ligno-cellulosic waste [37,38,39] with certain identified specific peaks located at 1727, 1516, 1370, 1157, and 1064 cm^−1^. The destructuration process of ligno-cellulosic compounds and the appearance of the graphitic configuration after the pyrolytic processing are highlighted by the significant changes occurring in the recorded spectrum of the CDots (Figure 3b). Thus, the most intense peak located at 1064 cm^−1^ due to C_6_−O_6_H stretching (cellulose), and C−O stretching (lignin) was no longer present. The most intense peak recorded in the case of the prepared CDs was located at 1611 cm^−1^ due to the stretching vibrations of the carbonyl groups, being specific to Carbon Dots. Both peaks specific to C=O vibrations appear shifted to lower wave numbers due to the reconfigurations that occurred during the thermal processing. The stretching vibrations specific to the C−H bond also appeared displaced and split in two peaks located at 2926 and 2859 cm^−1^ respectively, as a result of the establishment of a graphitic configuration. The recorded results also revealed the presence of various functional groups in the structure of the prepared CDots, which are responsible for the radiative transitions specific to these types of nanostructures.

### 3.4. Raman Spectroscopy

Figure 4 presents the Raman spectra recorded for the prepared CDs. The peaks recorded at 1376 and 1582 cm^−1^ are typical for graphitic structures resulting from the carbonization processes of organic materials used as precursors for the preparation of CDs [40,41].

The peak located at 1582 cm^−1^ (G band) corresponds to the in-plane stretching vibration in the E_2g_ mode of the graphitic structures, being due to sp^2^ carbon atoms. The peak located at 1376 cm^−1^ (D band) is specific to the swinging bond vibrations of the carbon atoms located at the periphery of the plane in the disordered graphitic structures, being an indicator of the structural defects. The intensity ratio of these two ID/IG peaks is proportional to the disorder degree or structural defects located in the graphitic structure, also indicating the ratio between sp^3^/sp^2^ carbon atoms [42]. In the case of CDs prepared from ligno-cellulosic waste, the ID/IG ratio was approx. 1/3, which suggests a relatively ordered graphitic structure. The structural defects most likely occurred due to the intercalation of various atomic species (e.g., oxygen) in the graphitic structure or the existence of certain functional groups attached to the terminal carbon atoms in the graphitic network. The recorded results are in agreement with those acquired through FT-IR investigation, which highlighted the existence of various functional groups within the structure of the CDs.

### 3.5. XRD Analysis

Figure 5 presents the XRD pattern recorded for the prepared CDs. The broad peak located at 28.51° 2θ is typical for graphitic structures [43]. The peak is slightly up-shifted compared with plain graphite peak (26.4° 2θ) corresponding to a minor decrease of the plane spacing between the carbon layers from 3.37 Å to 3.12 Å most probably due to the presence of various atomic species inserted within the graphitic structure as was also suggested by the Raman investigation results. The presence of the narrow peaks located at 39.49 and 45.71° 2θ are a result of the metallic residues (Section 3.1, EDX Investigation) inherently trapped within the graphitic core.

### 3.6. Dimensional Analysis DLS

For the dimensional analysis, the aqueous dispersed CDs sample was investigated as resulted after the final centrifugation stage (Cap. 2.3.), while the sample of acetone-dispersed CDs was prepared through an additional stage of freeze-drying and re-dispersion in acetone. It was observed that CDs dispersions had an agglomeration tendency, which may impact the PL emission intensity. This trend is most likely to occur due to the functional groups located on the surface of the graphitic core. Thus, the functional groups specific to an individual CDs entity interact with those of neighboring CDs, ultimately leading to their organization in clusters. In general, agglomeration occurs after a period of 48–72 h, and therefore CDs dispersions should be used freshly prepared. One solution to overcome the agglomeration is to embed the freshly prepared CDs in a polymer matrix, for example, Poly(vinyl alcohol) or Poly(vinyl pyrrolidone), which, after drying, will quench the agglomeration tendency and long term maintain the PL emission intensity. Figure 6 presents the size distributions (%) for the aqueous and acetone-dispersed CDs.

The recorded average sizes were 50 nm for the water dispersed CDs and 80 nm for those dispersed in acetone. The dimensional dispersion was relatively low, the predominant percentages being in the range of 40–48 nm in the case of the aqueous dispersion and 65–75 nm for the dispersion in acetone. In the case of acetone-dispersed CDs, the higher recorded average size is most probably a result of the additional clustering due to the required freeze-drying and re-dispersion stages. In both cases, the investigation most probably revealed already clustered CDs, the dimensions of an individual CD entity being much smaller.

### 3.7. HR-SEM Imaging

Figure 7a,b presents two HR-SEM micrographs recorded at different magnifications and observation fields. As could be noted, the entities are mostly within the 20–80 nm range with smaller structures being also visible, which is in very good agreement with the results recorded through DLS investigation. Their geometry is most probably a result of the agglomeration tendency since the observed entities are certainly not individual CDs but rather clusters with less regular shapes. As stated above, the agglomeration tendency due to the presence of surface-located functional groups might have a significant impact on the PL emission of the CDs. It is very likely that the clusters up to a certain dimension to favor the radiative processes [44] while the further agglomeration achieved after a longer time (48–72 h after dispersion) leads to an opposite effect.

### 3.8. Fluorescence Spectroscopy

Although the luminescent emission is one of the most interesting characteristics of CDs, the mechanisms involved in the radiative processes are not fully elucidated, remaining an open topic of debate. The luminescence mechanisms, according to the studies carried out so far, can be grouped into two main categories: one approach suggests the processes of quantum confinement occurring in nanometric structures, with a mechanism similar to the one involved in the fluorescent emission of semiconductor Quantum Dots nanocrystals. Another approach suggests mechanisms based on the radiative relaxation of excited states achieved by various species or functional groups within the CDs; such are the defects located in the graphitic core or various surface-located terminal groups. Additionally, excited states and radiative relaxations can appear in the functional groups located on the surface or can be due to the interactions between them [45]. Other studies [46,47] suggest the formation of intermediate organic fluorophores as a result of pyrolytic processing and partial thermal destruction of precursors. This approach only partially explains the particular emission characteristics of CDs with their emission peaks dependent on the excitation wavelength, this behavior being atypical for classic fluorophores. From the experimental data and extensive theoretical studies, the particular PL emission properties of CDs are most likely based on the radiative transitions occurring within/between the functional groups located on the surface of the graphitic core, the dimensional characteristics having a limited influence or an indirect role in the PL emission. In Figure 8 are presented the recorded PL spectra for the prepared CDs.

The emission spectra were recorded at excitation wavelengths within 350–390 nm range (step 10 nm), the emission peaks being detailed in Table 3. As could be noted, the prepared CDs present the characteristic excitation dependent emission peaks which are located in the blue–blue–green regions (438–473 nm) of the visible spectrum.

The most intense emission peak is located at 438 nm was obtained at an excitation wavelength of 350 nm. The Stokes shifts remained relatively constant throughout the excitation range, being within the limits of 78–88 nm, which indicates a comparable efficiency of the radiative processes. The absolute photoluminescence quantum yields (PLQY) were studied in the same excitation range, the highest value (28.4%) being obtained at 350 nm. The significant value of the recorded PLQY allows potential implementation in various applications ranging from sensors, contrast agents for biology investigations, to photonic conversion mediums in various optoelectronic devices (PV cells, PC-LEDs), having additional advantages related to the high availability of biomass waste, lack of toxicity, and easy preparation with low associated costs.

Upon excitation with a regular UV-A source, the dominant emission is located in the blue region of the visible spectrum. Figure 9a shows the chromatic parameters (according to CIE 1931) recorded at a 350 nm excitation wavelength (where the PLQY value is maximum).

Figure 9b presents the results of excited states lifetime investigation while Table 4 details the recorded values of each. The recorded lifetimes of the excited states are within nanosecond range, the values being typical for Carbon Dots-type nanostructures [48]. The recorded values support mechanisms resulting from radiative relaxations of the excited states as a result of excitation and interactions occurring within or between surface located functional groups and/or defects located in the graphitic core. A significant contribution could be due to the carbonyl groups attached to the surface or inside the graphitic core, which independently or in interaction with other vicinal groups could have a key role in achieving the PL emission.

## 4. Conclusions

Carbon Dots with intense blue photo-luminescent emission were prepared through pyrolytic processing of forestry ligno-cellulosic waste. The preparation path is simple and straightforward, mainly consisting of drying and fine-grinding of the ligno-cellulosic waste followed by thermal exposure and dispersion in water. The prepared Carbon Dots presented characteristic excitation wavelength-dependent emission peaks ranging within 438–473 nm and a remarkable 28% quantum yield achieved at 350 nm excitation wavelength. Morpho-structural investigations revealed the typical characteristics of the Carbon Dots class nanomaterials.

Due to the wide availability of this type of ligno-cellulosic waste, easy processing procedure, and achieved photo-luminescent properties, the prepared CDs could be an interesting approach for various applications ranging from sensors, contrast agents for biology investigations, to photonic conversion mediums in various optoelectronic devices. The prepared CDs could be easily be embedded in various polymer matrices, resulting in nanocomposites which could be easily tailored according to the intended application. Additionally, their biocompatibility and waste valorization in new materials might be equally good arguments in their favor, bringing a truly “green” approach.

## Figures and Tables

**Figure 1 nanomaterials-13-00131-f001:**
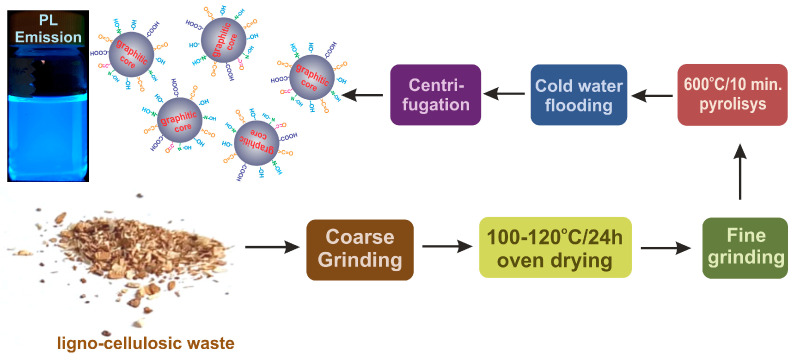
Preparation scheme of the photo emissive CDs from forestry ligno-cellulosic waste.

**Figure 2 nanomaterials-13-00131-f002:**
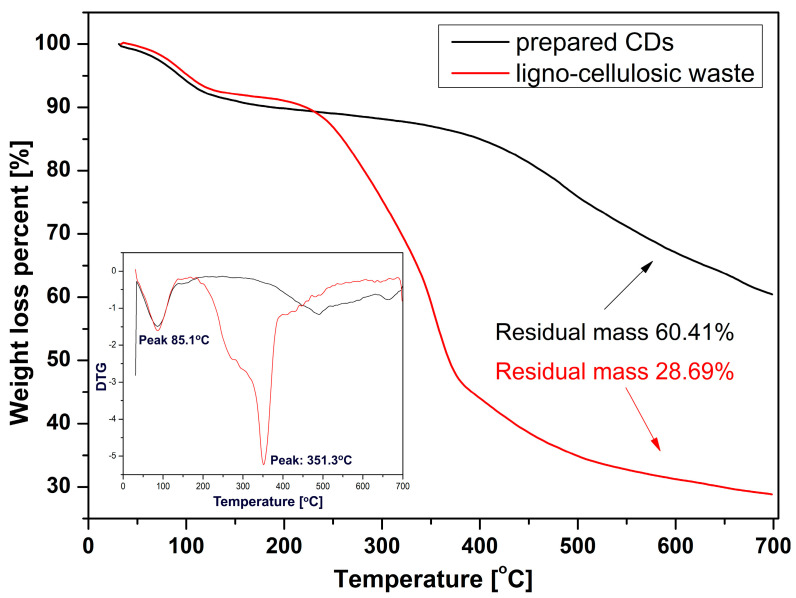
Mass variation recorded through thermal analysis for ligno-cellulosic waste and CDs samples.

**Figure 3 nanomaterials-13-00131-f003:**
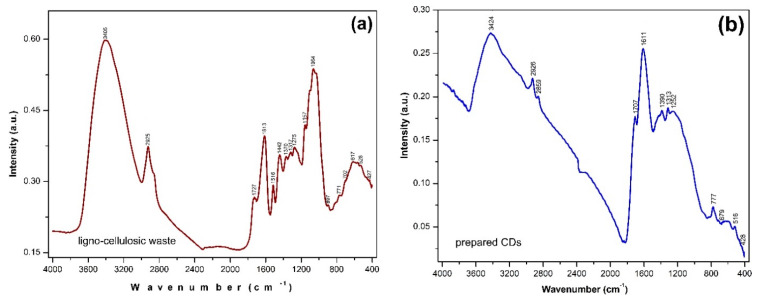
Recorded IR spectra for (**a**) the ligno−cellulosic waste and (**b**) the prepared CDs.

**Figure 4 nanomaterials-13-00131-f004:**
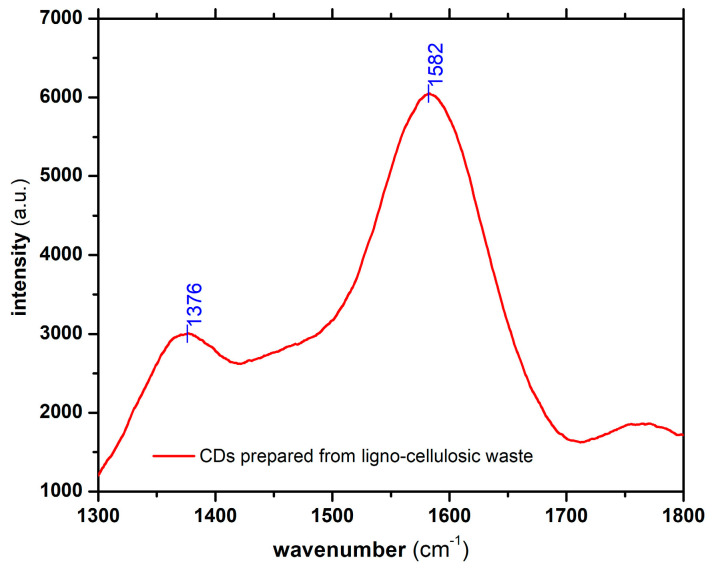
Recorded Raman spectrum for the prepared CDs.

**Figure 5 nanomaterials-13-00131-f005:**
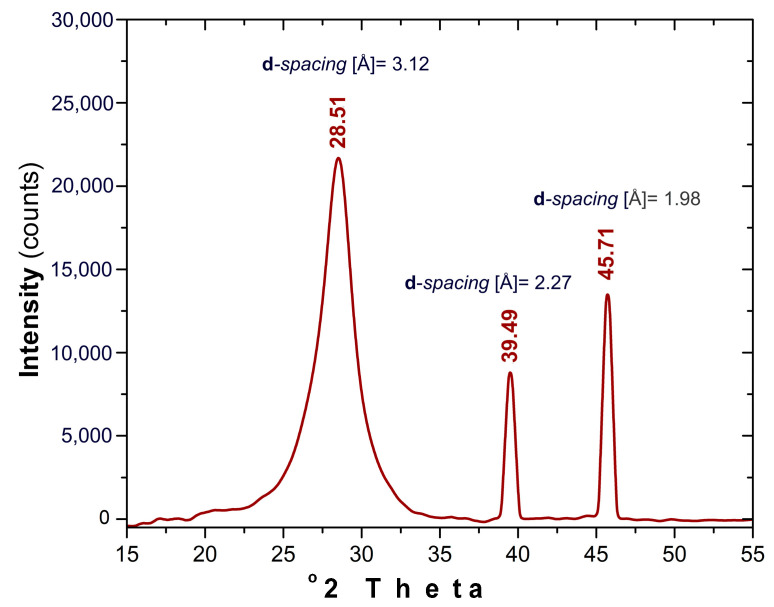
XRD pattern recorded for the prepared CDs.

**Figure 6 nanomaterials-13-00131-f006:**
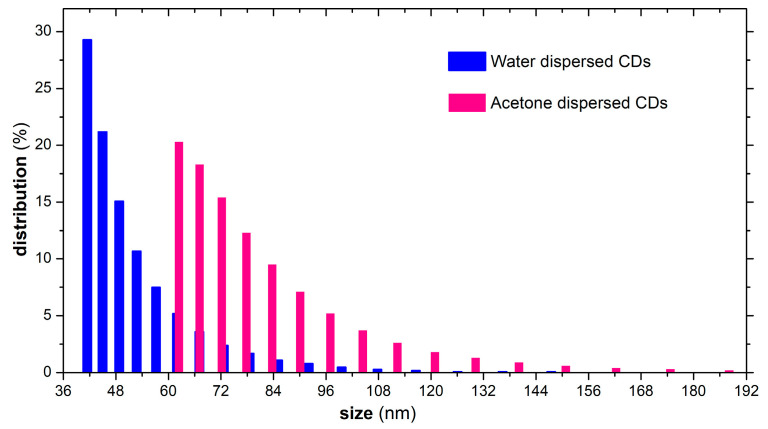
Dimensional distribution recorded for water and acetone dispersed CDs.

**Figure 7 nanomaterials-13-00131-f007:**
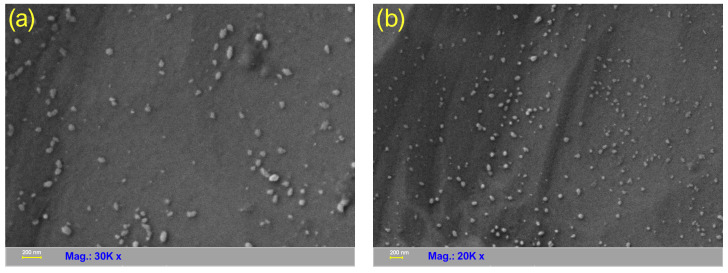
HR-SEM micrographs (**a**) 30 K magnification and (**b**) 20 K magnification recorded the prepared CDs.

**Figure 8 nanomaterials-13-00131-f008:**
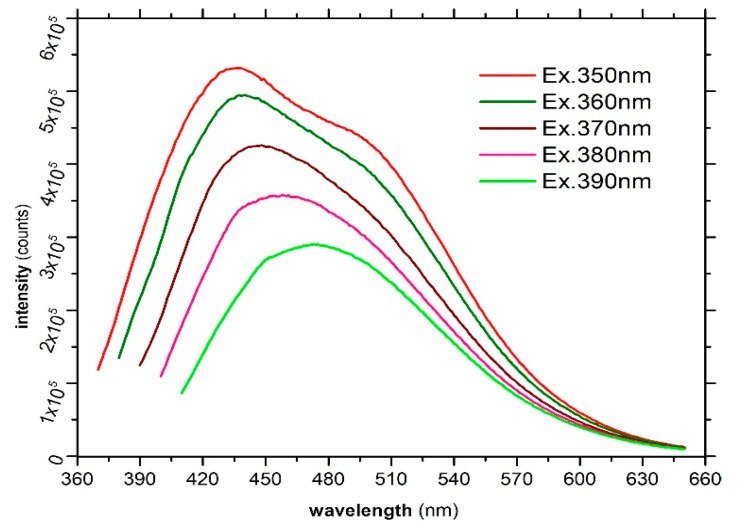
Steady-state PL emission spectra of the prepared CDs recorded within the 350–390 nm excitation range.

**Figure 9 nanomaterials-13-00131-f009:**
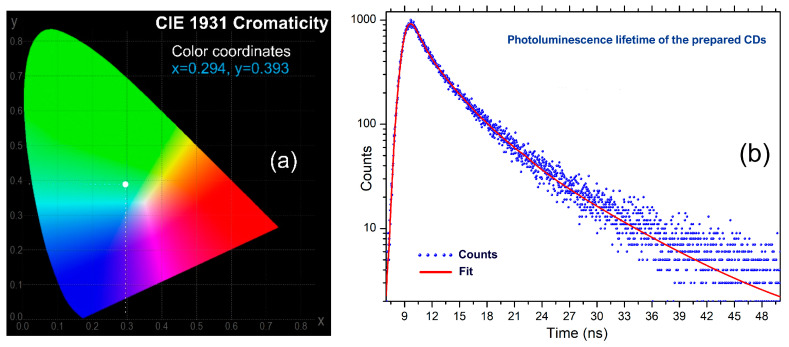
(**a**) CIE1931 chromaticity parameters and (**b**) PL Lifetime decay recorded for the prepared CDs.

**Table 1 nanomaterials-13-00131-t001:** Elemental composition of the forestry ligno-ellulosic waste.

Element	Weight %	Atomic %
C K	62.8	68.7
NK	6.8	6.4
O K	30.2	24.7
FeK	0.1	0.1
NiK	0.1	0.1

**Table 2 nanomaterials-13-00131-t002:** Significant peak assignment for the ligno-cellulosic waste.

Wavenumber (cm^−1^)	Assignement
3405	−OH stretching
2925	C−H stretching
1727	C=O stretching in acetyl and carboxyl groups
1613	C=O stretching in carbonyl groups
1516	Specific vibration of lignin aromatic ring
1442	O−H in plane bending, C-H bending
1516	Specific vibration of lignin aromatic ring
1442	O−H in plane bending, C-H bending
1370	C−H bending (cellulose, hemicellulose)
1317	O−H in plane bending (cellulose, hemicellulose)
1275	C-O stretching (lignin)
1157	C_1_−O−C_4_ anti-symetric bending (cellulose, hemicellulose)
1064	C_6_−O_6_H stretching (cellulose), C−O stretching (lignin)
897	C=C, C−H bending
771	C−H bending
528	C−OH out of plane bending

**Table 3 nanomaterials-13-00131-t003:** Emission peaks recorded within 350–390 nm excitation.

Excitation (nm)	Emission Peak (nm)
350	438
360	440
370	448
380	458
390	473

**Table 4 nanomaterials-13-00131-t004:** Recorded excited states’ lifetimes.

Lifetimes (ns)	Contribution (%)
*τ*_1_ = 2.585	46.86
*τ*_2_ = 0.501	20.90
*τ*_3_ = 8.337	32.23

## Data Availability

The data presented in this study are available on request from the corresponding author.

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
