# Peer review of "Intense Blue Photo Emissive Carbon Dots Prepared through Pyrolytic Processing of Ligno-Cellulosic Wastes"

_nanomaterials, 2022, doi:10.3390/nano13010131_

Round 1
Reviewer 1 Report
In this manuscript, the authors report the preparation of carbon dots from ligno-cellulosic wastes. The carbon dots were characterized by EDX, FT-IR, Raman spectra, and Dimensional analysis. The luminescent properties of the carbon dots, including emission spectra, luminescent quantum yield, and lifetime, were also studied. The idea of the reuse of waste is interesting, but there are still some major issues that should be considered before accepting the manuscript.
1. Mostly, the carbon dots should be characterized by TEM, SEM, and XPS, as shown in ref. 22, thus I would like to suggest the authors add these images to the revised manuscript.
2. I would like to suggest the authors provide the XRD pattern of the carbon dots.
3. There are many formatting errors in the manuscript, for example, some letters in equation 1 should be italic, so please check them carefully and standardize the format.
Author Response
Dear reviewer,
Many thanks for your useful and constructive comments. Please find our the point to point reply at reviewer’s comments in the attached pdf document. Please see the attachment.

Reviewer 2 Report
Stan et al reported the blue emissive carbon dots prepared through pyrolytic processing of ligno-cellulosic wastes. The preparation path is simple and straightforward mainly consisting of drying and fine grinding of the lingo-cellulosic waste followed by thermal exposure and dispersion in water. The prepared Carbon Dots present characteristic excitation wavelength dependent emission peaks ranging within 438-473 nm and a remarkable 28% quantum yield achieved at 350 nm excitation wavelength. The structure of prepared carbon dots was also investigated by the EDX, FT-IR, Raman, DLS in detail. The work was carefully carried out and the results are nice and significant. It will provide valuable addition to the design of the functional carbon dots. I recommend publication of this work in Nanomaterials after some revisions:
1. The quality of the figures presented in the manuscript can be improved.
2. English language should be improved, and spell check will be required and carefully checked.
3. Based on the structural characterization, the possible structural illustration of the prepared carbon dots should be provided in the manuscript.
4. Some important recent works on the functional carbon dots should be added for more balanced reference citation: (e.g., Acc. Mater. Res. 2022, 3, 319; Inorg. Chem. 2019, 58, 13394; Adv. Funct. Mater. 2020, 30, 1910530; Adv. Funct. Mater. 2019, 29, 1902466; Adv. Funct. Mater. 2018, 28, 17064, etc.).
Author Response
Dear reviewer,
Many thanks for your useful and constructive comments. Please find our point to point reply at reviewer’s comments in the attached pdf document. Please see the attachment.

Round 2
Reviewer 1 Report
The authors have made the necessary revisions, thus I suggest the acceptance of the manuscript as it is.